# Characteristics of a diagnostic bronchoscopy in hypersensitivity pneumonitis

**Kim C. Styrvoky**[1], **Kiran Batra**[2], **Mark Robertshaw**[1], **Margaret Kypreos**[1], **An Lu**[1], **Craig S. Glazer**[1], **Traci N. Adams**[1] *

**1** Division of Pulmonary and Critical Care Medicine, University of Texas Southwestern Medical Center, Dallas, TX, United States of America, **2** Department of Radiology, University of Texas Southwestern Medical Center, Dallas, TX, United States of America

* traci.adams@utsouthwestern.edu

## Abstract

### Background

Bronchoalveolar lavage and transbronchial biopsy can increase diagnostic confidence in the diagnosis of hypersensitivity pneumonitis (HP). Improving the yield of bronchoscopy may help to improve diagnostic confidence while decreasing the risk of potential adverse outcomes associated with more invasive procedures such as surgical lung biopsy. The purpose of this study is to identify factors that were associated with a diagnostic BAL or TBBx in HP.

### Methods

We conducted a retrospective cohort study of HP patients at a single center who underwent bronchoscopy during the diagnostic evaluation. Imaging characteristics, clinical characteristics including use of immunosuppressive medications and presence of active antigen exposure at the time of bronchoscopy, and procedural characteristics were collected. Univariable and multivariable analysis was performed.

### Results

88 patients were included in the study. 75 patients underwent BAL and 79 patients underwent TBBx. Patients who had an active fibrogenic exposure at the time of bronchoscopy had a higher BAL yield than those who were out of exposure at the time of bronchoscopy. TBBx yield was higher when more than 1 lobe was biopsied, with a trend toward higher yield of TBBx when nonfibrotic lung was biopsied compared to fibrotic lung.

### Discussion

Our study suggests characteristics that may improve yield of BAL and TBBx in patients with HP. We suggest that bronchoscopy be performed when patients are in the antigen exposure and that TBBx samples are taken from more than 1 lobe in order to improve diagnostic yield of the procedure.

author coding has also been uploaded as
Supplementary Information.

**Funding:** The authors received no specific funding
for this work.

**Competing interests:** The authors have declared
that no competing interests exist.

## Introduction

Bronchoalveolar lavage and transbronchial biopsy can increase diagnostic confidence in the diagnosis of hypersensitivity pneumonitis (HP) [1–3]. Bronchoscopy is particularly useful in patients with a high-resolution computed tomography (HRCT) that is not consistent with UIP, in those with identified antigen exposure, and in those who do not meet criteria for connective tissue disease related ILD [4, 5]. While sensitivity of transbronchial biopsy (TBBx) is much lower than surgical lung biopsy (SLB) due to smaller sample size, the complication rate of TBBx is less than that of SLB [4, 6–9]. Transbronchial lung cryobiopsy (TBLC) for the diagnosis of ILD is an emerging technique that may provide an alternative to SLB. European Respiratory Society guidelines suggest TBLC as a replacement test in patients eligible for SLB [10], and American Thoracic Society guidelines provide a conditional recommendation for TBLC as an alternative to SLB in medical centers with expertise in performing and interpreting TBLC results [11]. Unfortunately, TBLC is not available at all centers and is higher risk for complications than TBBX [12]. Therefore, appropriate patient selection for bronchoscopy and optimization of the procedure including standard BAL and TBBX may be helpful in improving diagnostic confidence while reducing the need for a higher risk procedures such as SLB.

Recent guidelines on the diagnosis of HP suggest that BAL improves diagnostic confidence for HP [3], and a prior study showed that the combination of bronchoalveolar lavage (BAL) and TBBx can improve diagnostic confidence and avoid SLB in up to 50% of patients with HP [13]. While patients are generally selected for bronchoscopy based on their pre-test probability of HP, it remains unclear which features of the procedure can optimize yield.

Improving the yield of bronchoscopy may help to improve diagnostic confidence while decreasing the risk of potential adverse outcomes associated with more invasive procedures such as SLB. Therefore, we sought to identify factors that were associated with a diagnostic BAL or TBBx in HP. We hypothesized that patients who remained in HP exposure would have a higher yield on bronchoscopy relative to those who removed the exposure prior to the procedure.

## Methods

We retrospectively identified HP patients evaluated between 2011–2019 from the University of Texas Southwestern Medical Center (UTSW). This study was conducted in accordance with the amended Declaration of Helsinki and was approved by the UTSW Institutional Review Board (STU-2019-0913). We are reporting a retrospective study of medical records, and the IRB waived requirement for informed consent. Patients who had a bronchoscopy for the diagnostic workup of their HP were included in the study. HP patients were included if they had a moderate, high, or definite probability of HP by the American Thoracic Society guidelines [3]. Patients were excluded if the bronchoscopy report was not available for review.

Clinical data extracted from the medical record included age, gender, smoking history, potential fibrogenic antigen exposure, date of exposure removal, pulmonary function testing (PFTs), BAL cell count and differential, histopathologic interpretation of the TBBx and SLB, number of lobes biopsied by TBBx, and use of immunosuppressive medications. HRCTs were evaluated by a thoracic radiologist (KB) [14]. Patients were considered to be out of antigen exposure at the time of bronchoscopy if the clinic notes indicated that the patient had removed the fibrogenic exposure at least 1 week prior to the procedure. Our antigen identification is done via a template questionnaire at the initial visit; the questionnaire is repeated if it is negative and the diagnostic evaluation suggests HP. We rarely perform serum precipitans testing except in cases in which the antigen exposure history is unclear or indeterminate.

BAL lymphocyte percentage > 30% was considered supportive of a diagnosis of hypersensitivity pneumonitis [3, 5, 15–18]. CD4/CD8 ratio is not routinely performed at our center and is not reported in this study, as it is not recommended in the current HP guidelines and has a recommendation against its routine use in bronchoalveolar lavage guidelines due to its poor sensitivity for HP, variability with age, and fluctuation during the course of illness [2, 3]. TBBx was considered characteristic of HP in patients with fibrosis if it had poorly formed non-necrotizing granulomas and either chronic fibrosing interstitial pneumonia or fibrotic nonspecific interstitial pneumonia like pattern [3, 19–21]. TBBx was considered characteristic of HP in nonfibrotic patients if it had poorly formed non-necrotizing granulomas, cellular interstitial pneumonia, and cellular bronchiolitis [3, 19–21].

## Statistical analysis

Means and standard deviations were used to express continuous variables, while Student's t test or Wilcoxon signed rank sum test were used to compare them. Counts and percentages were used to compare categorical variables, and Chi-squared test or Fisher's exact test were used to compare them. We used univariable logistic regression to identify factors that were associated with diagnostic BAL or TBBx. Prior literature indicates that patients with antigen exposure, those with an HRCT consistent with non-IPF diagnosis, and those without defined CTD led to an increased likelihood of diagnostic bronchoscopy in patients with ILD. In order to more definitively characterize the role of active antigen exposure and the role of radiographic findings in the areas sampled during bronchoscopy, whether the fibrogenic exposure had been removed prior to the bronchoscopy, number of lobes biopsied on TBBx, HRCT findings in the locations of BAL and TBBx, and use of immunosuppression as variables in the univariable analysis. In addition, we limited our sample to patients who were ultimately diagnosed with HP, rather than all patients with ILD, to remove potential confounding by indication. Variables that were significantly associated with change in diagnosis (p-value <0.2) were included in multivariable model to test independent associations. All p-values less than 0.05 were considered significant [5]. Statistical analyses were performed using MedCalc Statistical Software version 19.2.6 (MedCalc Software bv Ostend, Belgium; https://www.medcalc.org; 2020).

## Results

### Patient characteristics

In our retrospective cohort, 88 (100%) of patients had a moderate, high, or definite confidence of HP according to the American Thoracic Society guidelines and had undergone diagnostic bronchoscopy in the evaluation and were included in the analysis. Mean age at diagnosis was 62 years, and the cohort was primarily a non-Hispanic white population (Table 1). Bronchoscopy was performed in 88 patients (100%), with 85.2% having a BAL, 89.8% having a TBBx, and 75.0% having both BAL and TBBx. Our institutional practice is to obtain both BAL and TBBx in all patients unless the severity of illness of the patient precludes TBBx, in which case only BAL is performed. All of the patients who had TBBx but not BAL had the procedure performed outside of our institution by another provider. A potential fibrogenic antigen exposure was identified in 92% of the cohort; 25.3% of patients removed the exposure prior to bronchoscopy. Twelve patients (13.6%) were on immunosuppression at the time of bronchoscopy.

### BAL characteristics

Seventy-five patients (85.2%) underwent BAL. Median lymphocyte percentage was 25% (IQR 16.5–50.5), and 46.7% of patients had > 30% lymphocytes. In the univariable analysis, patients

**Table 1. Characteristics of HP cohort.**

| | HP cohort (N = 88) |
|---|---|
| Mean age (SD) | 61.9 (11.6) |
| Male, No. (%) | 42 (47.7) |
| Ethnicity, No. (%) | |
| Non-Hispanic White | 72 (81.8) |
| Black | 2 (2.3) |
| Hispanic or Latino | 7 (8.0) |
| Asian | 3 (3.4) |
| Other | 1 (1.1) |
| Unknown | 3 (3.4) |
| Ever Smoker, N (%) | 37 (42.0) |
| Antigen identified, No. (%)* | 81 (92.0) |
| Any Avian | 57 (64.8) |
| Mold | 37 (42.0) |
| Other | 11 (12.5) |
| Baseline Lung Function, mean (SD), N | |
| FVC % predicted | 69.5 (17.0) |
| DLCO % predicted | 54.7 (15.0) |
| HRCT available for scoring | 88 (100) |
| Consistent with a non-IPF diagnosis | 74 (84.1) |
| Indeterminate UIP | 9 (10.3) |
| Probable UIP | 2 (2.3) |
| Definite UIP | 3 (3.4) |
| Invasive procedure Performed** | 88 (100) |
| Surgical Biopsy | 30 (34.1) |
| TBBx | 79 (89.8) |
| BAL | 75 (85.2) |
| BAL and TBBx | 66 (75.0) |
| BAL lymph > 30% | 35 (46.7) |
| Tbbx diagnostic of HP | 34 (43.0) |
| On IS at time of bronch | 12 (13.6) |
| Out of exposure at time of bronch | 22 (25.0) |

*Some patients had more than one antigen identified

**Some patients underwent SLB and bronchoscopy

who had an active antigen exposure at the time of bronchoscopy had an odds ratio of 3.57 (95% CI 1.20–12.50, p = 0.028) for a BAL lymphocyte percentage > 30% (Table 2). Use of immunosuppression at the time of bronchoscopy and presence of ground glass, fibrosis, air trapping, and nodularity in the lobe of the BAL were not associated with BAL lymphocytosis. In a pre-specified multivariable analysis, only having an active antigen exposure at the time of bronchoscopy was associated with a higher yield (OR 3.33, 95% CI 1.09–11.11, p = 0.044). Mean BAL lymph count for patients who had removed exposure was 24.9%, compared to 38.0% in patients who were in exposure (p = 0.040).

## TBBx characteristics

Seventy-nine patients (89.8%) underwent TBBx. Thirty-four patients (43.0%) had histopathology from TBBx that supported a diagnosis of HP. Use of immunosuppression at the time of

**Table 2. Univariable and multivariable analysis of characteristics associated with BAL lymph > 30%.**

| | Univariable Analysis | | Multivariable Analysis | |
|---|---|---|---|---|
| | OR (95% CI) | p-value | OR (95% CI) | p-value |
| Ground glass | 1.92 (0.76–5.00) | 0.17 | 1.96 (0.75–5.26) | 0.18 |
| Fibrosis | 0.41 (0.13–1.17) | 0.10 | 0.45 (0.14–1.37) | 0.17 |
| Air trapping | 1.35 (0.51–3.70) | 0.55 | | |
| Nodularity | 0.91 (0.32–2.56) | 0.86 | | |
| Immunosuppression | 0.53 (0.11–2.19) | 0.40 | | |
| Active antigen exposure | 3.57 (1.20–12.50) | 0.028 | 3.33 (1.09–11.11) | 0.044 |

bronchoscopy, active antigen exposure at the time of bronchoscopy, and presence of ground glass, fibrosis, air trapping, and nodularity in the lobe of the TBBx were not associated with TBBx results that supported a diagnosis of HP (Table 3). In a pre-specified multivariable analysis, compared to TBBx occurring in 1 lobe of the lung, those with 2 (OR 4.33 95% CI 1.21–17.08, p = 0.028) or 3 (OR 3.91 95% CI 1.27–13.44, p = 0.022) lobes biopsied had higher yield. In the multivariable analysis, the presence of fibrosis in the lobe of the TBBx had a trend toward statistical significance for a lower yield of the procedure but did not reach statistical significance (OR 0.35, 95% CI 0.10–1.07, p = 0.07).

## Discussion

In this study, we evaluated the features of bronchoscopy that are positively associated with a BAL and TBBx supportive of a diagnosis of HP. Patients who had an active fibrogenic exposure at the time of bronchoscopy had a higher BAL yield than those who were out of exposure at the time of bronchoscopy. TBBx yield was higher when more than 1 lobe was biopsied, with a trend toward higher yield of TBBx when nonfibrotic lung was biopsied compared to fibrotic lung.

Our result builds on a prior study of the diagnostic yield of bronchoscopy in patients with ILD, which found that the identification of antigen, lack of a defined connective tissue disease, and non-IPF HRCT pattern increased diagnostic yield [5]. That study demonstrated that an increased number of features that are predictive of HP diagnosis will increase the yield of bronchoscopy, since bronchoscopy is useful in the diagnosis of HP but not in other ILDs such as IPF or CTD-ILD [3, 5, 11]. The present study was conducted in order to examine which features within a population of patients with HP improve diagnostic yield.

**Table 3. Univariable and multivariable analysis of characteristics associated with a diagnostic transbronchial biopsy.**

| | Univariable Analysis | | Multivariable Analysis | |
|---|---|---|---|---|
| | OR (95% CI) | p-value | OR (95% CI) | p-value |
| Ground glass | 1.33 (0.54–3.45) | 0.53 | | |
| Fibrosis | 0.52 (0.18–1.45) | 0.21 | 0.35 (0.10–1.07) | 0.07 |
| Air trapping | 0.88 (0.32–2.38) | 0.79 | | |
| Nodularity | 1.67 (0.61–4.64) | 0.32 | | |
| Immunosuppression | 0.87 (0.21–3.31) | 0.84 | | |
| Active antigen exposure | 0.68 (0.244–1.92) | 0.47 | | |
| # lobes sampled | | | | |
| 1 (ref) | REF | REF | REF | REF |
| 2 | 3.44 (1.02–12.30) | 0.049 | 4.33 (1.21–17.08) | 0.028 |
| 3 | 2.93 (1.026–8.92) | 0.049 | 3.91 (1.27–13.44) | 0.022 |

Improving the yield of bronchoscopy by appropriate patient selection and optimization of the procedure may help to improve diagnostic confidence while decreasing the risk of potential adverse outcomes associated with more invasive procedures such as SLB. In some HP patients, removal from exposure can be used as a diagnostic and therapeutic maneuver, where substantial improvement in FVC following exposure removal is considered diagnostic of HP [22]. In patients who fail to improve despite exposure removal, bronchoscopy may be used to improve diagnostic confidence [3, 22]. Our study suggests that bronchoscopy is higher yield when performed for patients who are in the exposure at the time of the procedure. Thus, it may be useful to perform bronchoscopy prior to exposure removal, particularly in patients who are unlikely to improve with exposure removal enough to make a definitive diagnosis, such as those with primarily fibrotic HP as opposed to a nonfibrotic phenotype [22]. Alternatively, a lower cutoff for BAL lymphocyte percentage could be used for patients who have removed antigen exposure, though larger studies would be needed to evaluate that strategy.

Our findings also demonstrate improved diagnostic yield when more than one lobe is biopsied and suggest improved diagnostic yield when nonfibrotic lung is biopsied. This fits with data from sarcoidosis which shows that increased number of samples lead to an increase in diagnostic yield [23]. Because the number of samples taken were inconsistently reported, we used number of lobes biopsied as a proxy for number of samples taken.

Strengths of our study include the use of consensus criteria to define a diagnosis of HP, thus limiting incorporation bias [3]. Bronchoscopies were conducted both at our center and by outside providers, improving generalizability.

There are several limitations to our study. Due to the retrospective nature of the study, some patients with HP at our center have not undergone BAL and/or TBBx. This study evaluated patients who underwent conventional BAL and TBBX, so the emerging usage of TBLC with potential for larger tissue samples may further influence decisions regarding initial diagnostic procedure. Given the retrospective nature of our study, we cannot definitively confirm compliance with immunosuppression at the time of bronchoscopy. Additionally, an industrial hygienist was not used to confirm the presence of antigen exposure or exposure removal; while this may limit our ability to definitively determine whether a patient had active exposure at the time of bronchsocopy, given lack of access to industrial hygienists in most centers, this pragmatic study remains generalizable to other ILD centers in that patient history is used to evaluate for exposure removal. Finally, our high rate of antigen detection likely represents confounding by indication. Patients with an identified antigen were more likely to undergo bronchoscopy than patients without identified antigen.

## Conclusion

In summary, current guidelines suggest that bronchoscopy may be useful in improving diagnostic confidence in patients with HP and is a lower risk procedure than SLB [3]. Here, we identified specific procedural features that help to improve yield for bronchoscopy in HP. We suggest that bronchoscopy is higher yield when performed while the patient is in the exposure and when TBBx are performed in more than one lobe.

## Supporting information

**S1 File. Manuscript data.** De-identified patient-level data from our cohort.
(XLSX)

**S2 File. Key to author coding.** Key to all author coding contained in S1 File.
(XLSX)

## Author Contributions

**Conceptualization:** Kim C. Styrvoky, Kiran Batra, Mark Robertshaw, Margaret Kypreos, An Lu, Craig S. Glazer, Traci N. Adams.

**Data curation:** Kim C. Styrvoky, Kiran Batra, Mark Robertshaw, Margaret Kypreos, An Lu, Craig S. Glazer, Traci N. Adams.

**Formal analysis:** Kim C. Styrvoky, Traci N. Adams.

**Investigation:** Kim C. Styrvoky, Traci N. Adams.

**Methodology:** Margaret Kypreos, Craig S. Glazer, Traci N. Adams.

**Writing – original draft:** Kim C. Styrvoky, Kiran Batra, Mark Robertshaw, Margaret Kypreos, An Lu, Craig S. Glazer, Traci N. Adams.

**Writing – review & editing:** Kim C. Styrvoky, Kiran Batra, Mark Robertshaw, Margaret Kypreos, An Lu, Craig S. Glazer, Traci N. Adams.

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
