## [Decision Letter · Decision Letter 0]

24 Apr 2023

PONE-D-23-05395Characteristics of a diagnostic bronchoscopy in hypersensitivity pneumonitisPLOS ONE

Dear Dr. Adams,

Thank you for submitting your manuscript to PLOS ONE. After careful consideration, we feel that it has merit but does not fully meet PLOS ONE’s publication criteria as it currently stands. Therefore, we invite you to submit a revised version of the manuscript that addresses the minor points raised during the review process.

We look forward to receiving your revised manuscript.

Kind regards,

Bintou Ahidjo

Academic Editor

PLOS ONE

Journal Requirements:

4. Your ethics statement should only appear in the Methods section of your manuscript. If your ethics statement is written in any section besides the Methods, please move it to the Methods section and delete it from any other section. 

Reviewers' comments:

Reviewer's Responses to Questions

**Comments to the Author**

1. Is the manuscript technically sound, and do the data support the conclusions?

Reviewer #1: Yes

2. Has the statistical analysis been performed appropriately and rigorously? 

Reviewer #1: Yes

3. Have the authors made all data underlying the findings in their manuscript fully available?

Reviewer #1: Yes

4. Is the manuscript presented in an intelligible fashion and written in standard English?

Reviewer #1: Yes

5. Review Comments to the Author

Reviewer #1: This is a nice retrospective CS on determinants of a diagnostic utility of bronchoscopy in patiens with HP. The results and conclusions are not suprising. However, I think it would be worth to publish these data as the paper is very well writen und straight forward. However, there are some issues, which have to be sold in a revised version:

1) Introduction, 1st para: "TBC .. has not yet established a defined role in ILD diagnostic algorithm". Please, add information that TBLC (which is the usual abbreviation) is suggested as a replacement test in patients considered eligible to undergo SLB (

Eur Respir J 2022 Nov 10;60(5):2200425. ) and holds conditional recommendation according to ATS/ERS recommendation (Am J Respir Crit Care Med. 2022 May 1;205(9):e18-e47.)

2) Results, 1st para: Please, explain on which findings the diagnosis of HP was established in the 88 patients? Eventually, how sure can we be that the included patients had HP and not another diagnosis?

3) Results, 1st para: 85.2% had BAL and 89.8% had TBBx. This is very unusual. Why was proportion of patients with BAL als compared to TBBx? The opposite would have been obvious.

4) Results, 1st para: 92% of patients had an identifiable antigen exposure. I am impressed, since this proportion is usually lower. Was antigen exposure only based patient's history or blood tests? How do authors explain this unusually high rate of positive findings?

5) Results, 2nd para: 86.2% of patients had BAL, but in the 1st para the proportion was 85.2%. This is confusion. Please, explain.

6) Results, 2nd para: There is no information on CD4/CD8 coefficient. Please, add this information, since this is an important issue when interpretin BAL cell count and differential. Why was CD4/CD8 coefficient not included in statistical analysis? In case it was not available (reason?), this major limitation should be added and discussed approprietly!

7) Results, 3rd para: 90.8% of patients had TBBx, but in the 1st para the proportion was different. Please, explain.

6. PLOS authors have the option to publish the peer review history of their article (what does this mean?). If published, this will include your full peer review and any attached files.

Reviewer #1: **Yes: **Daniel Franzen

---

## [Author Response · Author response to Decision Letter 0]

25 Apr 2023

We appreciate the reviewer’s comments. Please see our responses below.

We have updated the naming of the files, heading size and font, and references to tables within the text.

We have added the following clarification statement.

“We are reporting a retrospective study of medical records, and the IRB waived requirement for informed consent.”

We have uploaded the key to the author code in a separate file that is clearly labeled and have noted this in the data accessibility portion of the manuscript submission website.

4. Your ethics statement should only appear in the Methods section of your manuscript. If your ethics statement is written in any section besides the Methods, please move it to the Methods section and delete it from any other section. 

Our ethics statement is included in the Methods section:

“This study was conducted in accordance with the amended Declaration of Helsinki and was approved by the UTSW Institutional Review Board (STU-2019-0913). We are reporting a retrospective study of medical records, and the IRB waived requirement for informed consent.”

We have added a caption for the supporting information files.

We have reviewed the references. None have been retracted.

Reviewers' comments:

Comments to the Author

Reviewer #1: This is a nice retrospective CS on determinants of a diagnostic utility of bronchoscopy in patiens with HP. The results and conclusions are not suprising. However, I think it would be worth to publish these data as the paper is very well writen und straight forward. However, there are some issues, which have to be sold in a revised version:

1) Introduction, 1st para: "TBC .. has not yet established a defined role in ILD diagnostic algorithm". Please, add information that TBLC (which is the usual abbreviation) is suggested as a replacement test in patients considered eligible to undergo SLB (Eur Respir J 2022 Nov 10;60(5):2200425. ) and holds conditional recommendation according to ATS/ERS recommendation (Am J Respir Crit Care Med. 2022 May 1;205(9):e18-e47.)

We added these references to the introduction and added more text to explain the role of TBLC outlined in the ATS and ERS guidelines. We also corrected the abbreviation when used in the discussion section from TBC to TBLC.

“Transbronchial lung cryobiopsy (TBLC) for the diagnosis of ILD is an emerging technique that may provide an alternative to SLB. European Respiratory Society guidelines suggest TBLC as a replacement test in patients eligible for SLB,[9] and American Thoracic Society guidelines provide a conditional recommendation for TBLC as an alternative to SLB in medical centers with expertise in performing and interpreting TBLC results.[10] Unfortunately, TBLC is not available at all centers and is higher risk for complications than TBBX.[11]”

2) Results, 1st para: Please, explain on which findings the diagnosis of HP was established in the 88 patients? Eventually, how sure can we be that the included patients had HP and not another diagnosis?

These patients were diagnosed based on the ATS criteria for the diagnosis of HP, which takes into account antigen identification, HRCT findings, BAL, and histopathology. Each of these patients had a moderate, high, or definite confidence diagnosis by ATS guidelines. We have added this to the Methods section in the inclusion criteria and to the first paragraph of the results section to clarify.

To the Methods section we added: “HP patients were included if they had a moderate, high, or definite probability of HP by the American Thoracic Society guidelines.”

To the results section we added: “In our retrospective cohort, 88 (100%) of patients had a moderate, high, or definite confidence of HP according to the American Thoracic Society guidelines and had undergone diagnostic bronchoscopy in the evaluation and were included in the analysis.”

3) Results, 1st para: 85.2% had BAL and 89.8% had TBBx. This is very unusual. Why was proportion of patients with BAL als compared to TBBx? The opposite would have been obvious.

We agree that these findings are surprising. Our own institutional practice is to perform both BAL and TBBx when assessing for HP if able; when patients are too ill for TBBx, we perform BAL only. However, many of our patients underwent bronchoscopy by outside providers prior to referral to our ILD clinic. Unfortunately, most of these providers perform TBBx instead of BAL. We have done our best to provide education to community pulmonologist on the importance of BAL cell count and differential, but this is not uniformly done. 

We added a line to this paragraph to explain the above.

“Our institutional practice is to obtain both BAL and TBBx in all patients unless the severity of illness of the patient precludes TBBx, in which case only BAL is performed. All of the patients who had TBBx but not BAL had the procedure performed outside of our institution by another provider.”

4) Results, 1st para: 92% of patients had an identifiable antigen exposure. I am impressed, since this proportion is usually lower. Was antigen exposure only based patient's history or blood tests? How do authors explain this unusually high rate of positive findings?

Our antigen identification is done via a template questionnaire at the initial visit; the questionnaire is repeated if it is negative and the diagnostic evaluation suggests HP. We rarely perform serum precipitans testing except in cases in which the antigen exposure history is unclear or indeterminate. In this case, our antigen identification is likely quite high because antigen identification often led to the decision to perform bronchoscopy. We agree that we need to address this in the manuscript.

We have added the following to the Methods section: “Our antigen identification is done via a template questionnaire at the initial visit; the questionnaire is repeated if it is negative and the diagnostic evaluation suggests HP. We rarely perform serum precipitans testing except in cases in which the antigen exposure history is unclear or indeterminate.”

We added the following to our discussion section under limitations: “Our high rate of antigen detection likely represents confounding by indication. Patients with an identified antigen were more likely to undergo bronchoscopy than patients without identified antigen.”

5) Results, 2nd para: 86.2% of patients had BAL, but in the 1st para the proportion was 85.2%. This is confusion. Please, explain.

We sincerely apologize. This was a typographical error. The correct percentage of those undergoing BAL is 85.2%, and this has been corrected in the text. It is also available in Table 1.

6) Results, 2nd para: There is no information on CD4/CD8 coefficient. Please, add this information, since this is an important issue when interpretin BAL cell count and differential. Why was CD4/CD8 coefficient not included in statistical analysis? In case it was not available (reason?), this major limitation should be added and discussed approprietly!

We do not routinely perform CD4/CD8 ratio. This test is not recommended in the ATS guidelines for the diagnosis of HP in adults, the ATS guidelines for IPF, or the ATS guidelines for diagnosis of HP. We do, however, perform it when we are specifically evaluating for sarcoidosis, which was not typically high on the differential for these patients with antigen exposure, centrilobular nodules, and air trapping. A low CD4/CD8 ratio may be seen in HP, drug-induced lung disease, COP, eosinophilic pneumonia, and IPF, and the ratio has not been shown to contribute meaningfully to the diagnosis of non-sarcoid ILD as it cannot distinguish between these entities. If we were to perform prospective evaluation of BAL, I do think this ratio would be useful in order to provide clarity and transparency and possibly to outline HP patients in which this ratio is useful for prognosis or treatment response, but it is not performed in our suspected HP cases and therefore our data is not available.

We have added a statement to the Methods section: “CD4/CD8 ratio is not routinely performed at our center and is not reported in this study, as it is not recommended in the current HP guidelines and has a recommendation against its routine use in bronchoalveolar lavage guidelines due to its poor sensitivity for HP, variability with age, and fluctuation during the course of illness.[2, 3]”

7) Results, 3rd para: 90.8% of patients had TBBx, but in the 1st para the proportion was different. Please, explain.

We sincerely apologize for the oversight. The percentage has been corrected to 89.8% to match the first paragraph of the results section as well as Table 1.

---

## [Editor Report · Decision Letter 1]

27 Apr 2023

Characteristics of a diagnostic bronchoscopy in hypersensitivity pneumonitis

PONE-D-23-05395R1

Dear Dr. Adams,

We’re pleased to inform you that your manuscript has been judged scientifically suitable for publication and will be formally accepted for publication once it meets all outstanding technical requirements.

Kind regards,

Bintou Ahidjo

Academic Editor

PLOS ONE
---

## [Editor Report · Acceptance letter]

8 May 2023

PONE-D-23-05395R1 

Characteristics of a diagnostic bronchoscopy in hypersensitivity pneumonitis 

Dear Dr. Adams:

I'm pleased to inform you that your manuscript has been deemed suitable for publication in PLOS ONE. Congratulations! Your manuscript is now with our production department. 

Kind regards, 

on behalf of

Dr. Bintou Ahidjo 

Academic Editor

PLOS ONE